# Nonclassical steering and the Gaussian steering triangoloids

Massimo Frigerio[1,2], Stefano Olivares [2,3], Matteo G. A. Paris[2,3,*]

**1** Dipartimento di Fisica *Giuseppe Occhialini* dell'Università degli Studi di Milano-Bicocca, I-20126 Milano, Italy
**2** Dipartimento di Fisica *Aldo Pontremoli* dell'Università degli Studi di Milano, I-20133 Milano, Italy
**3** I.N.F.N. Sezione di Milano, I-20133 Milano, Italy
* matteo.paris@fisica.unimi.it

September 29, 2020

## Abstract

**Nonclassicality according to the singularity or negativity of the Glauber P-function is a powerful resource in quantum information, with relevant implications in quantum optics. In a Gaussian setting, and for a system of two modes, we explore how P-nonclassicality may be conditionally generated or influenced on one mode by Gaussian measurements on the other mode. Starting from the class of two-mode squeezed thermal states (TMST), we introduce the notion of *nonclassical steering* (NS) and the graphical tool of Gaussian triangoloids. In particular, we derive a necessary and sufficient condition for a TMST to be nonclassically steerable, and show that entanglement is only necessary. We also apply our criterion to noisy propagation of a twin-beam state, and evaluate the time after which NS is no longer achievable. We then generalize the notion of NS to the full set of Gaussian states of two modes, and recognize that it may occur in a *weak form*, which does not imply entanglement, and in a *strong form* that implies EPR-steerability and, a fortiori, also entanglement. These two types of NS coincide exactly for TMSTs, and they merge with the previously known notion of EPR steering. By the same token, we recognize a new operational interpretation of P-nonclassicality: it is the distinctive property that allows one-party entanglement verification on TMSTs.**

# 1   Introduction

The upsetting consequences of quantum correlations have attracted theoretical efforts since the early days of quantum mechanics. The first authors to pinpoint the new features of these correlations were probably Einstein, Podolsky and Rosen [1]. They considered two parties, that we will call Alice and Bob for our convenience, sharing a pair of quantum systems described altogether by the entangled pure state:

$$|\psi\rangle\rangle \;\; = \;\; \sum_a c_a |a\rangle_1 \otimes |a\rangle_2 \;\; = \;\; \sum_\alpha d_\alpha |\alpha\rangle_1 \otimes |\alpha\rangle_2$$

where $\{|a\rangle_j\}$ and $\{|\alpha\rangle_j\}$ are two distinct, orthonormal bases of the $j$-th system, $j = 1, 2$. They noted that, according to quantum mechanics, Alice can choose to perform a projective measurement on her system either in the $\{|a\rangle_1\}$ basis or in the $\{|\alpha\rangle_1\}$ basis, thereby collapsing the state of Bob's system into distinguishable quantum states (in $\{|a\rangle_2\}$ or in $\{|\alpha\rangle_2\}$, respectively), a phenomenon later termed *steering* by E. Schrödinger [2]. Refusing to abandon their notion of locality and observing that $\{|a\rangle_j\}$ and $\{|\alpha\rangle_j\}$, $j = 1, 2$ may be chosen to be the eigenstates of two non-commuting observables, EPR concluded that there was more to be known on Bob's system than what was provided by the quantum state's description [1]. In other words, they concluded that the quantum mechanical description of a physical system must be incomplete, if locality holds at the level of quantum states. The modern point of view is that locality does not hold for quantum states, but in such a way that *causality*, namely signal-locality, which is also the actual essential ingredient to special relativity, is still preserved. In fact, Bob cannot detect on his own the influence produced on his quantum state, in compliance with the no-signaling Theorem [3–6]. On the other hand, if Alice communicates her choice of measurement to Bob and they repeat the experiment many times, starting with the same entangled state each time, Bob can now check that Alice was indeed able to *steer* his state.

---

[1]We shall mean 'the quantum state of both systems' here, since we now know that the quantum state of just the second system is even less informative.

Since, for bipartite pure states, being entangled is precisely equivalent to being unfactorizable, we conclude that *quantum steering* is possible with a given pure bipartite state *if and only if* it is entangled.

At a variance with the pure case, mixed states can exhibit fully classical correlations, thus failing to be factorized even without entanglement. A generalized definition of entanglement for them was provided in [7], and it is based on the impossibility to construct an entangled state starting from a factorized one and using just local operations and classical communication (LOCC). There, the author also considered another celebrated manifestation of quantum correlations, the violation of Bell's inequality [8, 9], which was known to be possible with all and only entangled states in the pure case; Ref. [7] showed that among mixed states, instead, only a strict subset of entangled states allows for a violation of such inequality. It started to become apparent, then, that there is a true hierarchy of quantum correlations.

On this line, but only much later, the concept of quantum steering received a general formulation in [10, 11]. Since classical correlations in bipartite mixed states can mock the influence on one party by measurements on the other one, the idea of the authors was to declare that, given a shared bipartite state, Alice can steer Bob's state if she can condition his quantum state into different ensembles, in such a way that he cannot explain such an influence using a local hidden-variable model and assuming just classical correlations with Alice, and therefore he becomes convinced that the shared state was entangled. If there is a choice of quantum measurements on her party allowing Alice to convince Bob that the shared state was entangled, the state is called *steerable by Alice*. Contrary to the pure case, steering becomes a truly asymmetric property for mixed states, and it was shown [11] that one-way steerability is a stronger condition than entanglement, but a weaker condition than violation of Bell's inequality in general. The definition of quantum steering was also specialized to the case of Gaussian states of continuous-variable (CV) quantum systems, and we will refer to this notion as *EPR steering*. Steering is now widely considered a fundamental resource for quantum information tasks [12–17], for example in one-sided device-independent quantum key distribution (QKD) [12], and many criteria for its detection have been explored [18–21].

In addition to quantum correlations, a wealth of other concepts concerning the nonclassical character of quantum states have been put forward [22]. The nonclassicality of a CV quantum state $\hat{\boldsymbol{\rho}}$ is often determined by the behaviour of its Glauber P-function [23–26], which amounts to its expansion onto coherent states $|\alpha\rangle$ ($\alpha \in \mathbb{C}$) according to:

$$\hat{\boldsymbol{\rho}} \;=\; \int_{\mathbb{C}} \mathrm{d}^2\alpha \; P\left[\hat{\boldsymbol{\rho}}\right](\alpha) |\alpha\rangle\langle\alpha| \tag{1}$$

A major reason for the wide use of the P-function stems from its connection with experimentally accessible quantities, so that it leads to the most *physically inspired* notion of nonclassicality. It is known to be necessary for antibunching and sub-Poissonian photon statistics [27] in quantum optics, among others, whereas being *classical* according to the P-function implies the empirical adequacy of Maxwell's Equations in the phenomenological description of the corresponding state of light. On a practical level, the more nonclassical a state the harder to fabricate it with optical equipment [28, 29] (as in the case of highly squeezed states), and we may therefore say that P-nonclassicality has a *resource* character [30–32].

In this article, we shall explore the possibility of *steering* nonclassicality, in the setting of two-mode Gaussian states. In particular, we focus on Gaussian measurements on one of the modes and examine the effects on the nonclassicality of the state of the other mode, conditioned on the outcome of the measurement.

This paper is structured as follows. In Section 2 we set our notation for Gaussian states and P-nonclassicality. In Section 3 we consider the case of two-mode squeezed thermal states (TMSTs), introducing the concept of *nonclassical steering* and studying its relation with entanglement and its asymmetric behaviour. We also introduce *triangoloid plots*, a graphical tool that will guide our analysis and lead us to anticipate one of the main results: projective measurements on field quadratures are optimal, among all Gaussian measurements, to remotely influence nonclassicality. In Section 4 we discuss the situation of a twin-beam state (TWB), whose mode we wish to steer interacts with a noisy, thermal environment. We show that the resulting state is a generic TMST and we derive the maximum propagation time after which nonclassical steering is no longer viable, comparing it to the time needed to destroy all initial entanglement. In Section 5 we generalize our results to all Gaussian states of two modes, explaining the necessity to distinguish between *weak* and *strong* nonclassical steering. In particular, weak nonclassical steering will be seen to be independent on entanglement, therefore we examine its relation with Gaussian quantum discord. We conclude with a comparison between nonclassical steering and EPR steering, showing that they coincide for TMST states, with possible practical implications in quantum key distribution.

## 2 Gaussian states and nonclassicality

### 2.1 Phase space formalism and notation for Gaussian states

We consider the Fock space $\mathcal{F}^{(n)} = \bigotimes_{k=1}^{n} \mathcal{F}_k$ that is constructed as the tensor product of $n$ single-mode Fock spaces $\mathcal{F}_k$, each generated by the usual creation operators $\hat{a}_k^\dagger$ acting on the vacuum $|0\rangle_k$ of the respective mode, such that together with the corresponding annihilation operators $\hat{a}_k$ they satisfy the standard commutation relations for bosons. From these, we can define the ordinary conjugated canonical variables:

$$\hat{q}_k \ := \ \frac{\hat{a}_k + \hat{a}_k^\dagger}{\sqrt{2}} \ , \qquad \hat{p}_k \ := \ \frac{\hat{a}_k - \hat{a}_k^\dagger}{i\sqrt{2}}$$

which can be collected in the *quadrature vector* $\hat{\mathbf{R}} = (\hat{q}_1, \hat{p}_1, ..., \hat{q}_n, \hat{p}_n)^T$, so that the canonical commutation relations are compactly written as:

$$[\hat{R}_j, \hat{R}_k] \ = \ i\Omega_{jk} \tag{2}$$

$$\mathbf{\Omega} \ := \ \bigoplus_{k=1}^{n} \boldsymbol{\omega} \ , \qquad \boldsymbol{\omega} \ := \ \begin{pmatrix} 0 & 1 \\ -1 & 0 \end{pmatrix} \tag{3}$$

Given a state $\hat{\boldsymbol{\rho}}$, i.e. a trace-class, positive semidefinite, bounded linear operator from $\mathcal{F}^{(n)}$ to itself, we define its *characteristic function* [33, 34] as:

$$\chi[\hat{\boldsymbol{\rho}}](\Lambda) \ := \ \mathrm{Tr}[\hat{\boldsymbol{\rho}} \, \hat{\mathbf{D}}(\Lambda)] \tag{4}$$

where we introduced the *displacement operator*:

$$\hat{\mathbf{D}}(\Lambda) \ := \ \exp[-i\Lambda^T \mathbf{\Omega} \hat{\mathbf{R}}] \ = \ \bigotimes_{k=1}^{n} e^{\lambda_k \hat{a}_k^\dagger - \lambda_k^* \hat{a}_k} \tag{5}$$

with $\Lambda = (a_1, b_1, ..., a_n, b_n)^T$ and $\lambda_k = \frac{1}{\sqrt{2}}(a_k + ib_k)$. The Wigner function can be expressed in terms of the characteristic function as:

$$W[\hat{\boldsymbol{\rho}}](X) = \int_{\mathbb{R}^{2n}} \frac{\mathrm{d}^{2n}\Lambda}{(2\pi^2)^n} e^{i\Lambda^T \boldsymbol{\Omega} X} \chi[\hat{\boldsymbol{\rho}}](\Lambda) \tag{6}$$

We say that $\hat{\boldsymbol{\rho}}$ is a *Gaussian state* of $n$ modes if its Wigner function is a Gaussian function [2] on a $2n$-dimensional phase space, namely:

$$W[\hat{\boldsymbol{\rho}}](X) = \frac{1}{\pi^n \sqrt{\det[\boldsymbol{\sigma}]}} e^{-\frac{1}{2}(X - \langle \hat{\boldsymbol{R}} \rangle)^T \boldsymbol{\sigma}^{-1}(X - \langle \hat{\boldsymbol{R}} \rangle)} \tag{7}$$

where $\langle \hat{\boldsymbol{R}} \rangle = \mathrm{Tr}[\hat{\boldsymbol{\rho}} \hat{\boldsymbol{R}}]$ is the *first-moments vector* and:

$$[\boldsymbol{\sigma}]_{jk} = \frac{1}{2} \langle \hat{R}_j \hat{R}_k + \hat{R}_k \hat{R}_j \rangle - \langle \hat{R}_j \rangle \langle \hat{R}_k \rangle \tag{8}$$

is the *covariance matrix* (CM) of the state, and it is a positive semidefinite, symmetric matrix. Moreover, since it encodes the covariances for the expectation values of conjugate canonical observables, $\boldsymbol{\sigma}$ should fulfill the uncertainty relations (UR) that are valid for any physical state $\hat{\boldsymbol{\rho}}$, and they take the following form on phase space [35]:

$$\boldsymbol{\sigma} + \frac{i}{2}\boldsymbol{\Omega} \geq 0 \tag{9}$$

It is important to stress that Ineq. (9) is automatically true for any $\boldsymbol{\sigma}$ derived from the Wigner function of a Gaussian state $\hat{\boldsymbol{\rho}}$, whereas it has to be imposed on $\boldsymbol{\sigma}$ if a *physical* Gaussian state $\hat{\boldsymbol{\rho}}$ has to be defined from its Wigner function. In that case, any Gaussian function whose CM $\boldsymbol{\sigma}$ is symmetric, with $\boldsymbol{\sigma} \geq 0$ and fulfilling Ineq. (9), is the Wigner function of some physical Gaussian state $\hat{\boldsymbol{\rho}}$.

## 2.2 Nonclassicality

The Glauber P-function introduced before is a member of a continuous family of phase space quasiprobability distributions, known as *s*-ordered Wigner functions and defined according to [25, 36]:

$$W_s[\hat{\boldsymbol{\rho}}](X) = \int_{\mathbb{R}^n} \frac{d^{2n}\Lambda}{(2\pi^2)^n} e^{\frac{1}{4}s|\Lambda|^2 + i\Lambda^T \boldsymbol{\Omega} X} \chi[\hat{\boldsymbol{\rho}}](\Lambda) \tag{10}$$

for $s \in [-1, 1]$. With $s = 0$ we recover the Wigner function. Instead, the case $s = 1$, which is the most singular of the family and can behave even more singularly than a tempered distribution, corresponds precisely to the P-function. A CV quantum state $\hat{\boldsymbol{\rho}}$ is termed *nonclassical* [27, 37, 38] whenever its P-function is not positive semidefinite [39] and/or it is more singular than a delta distribution. One can also introduce the so-called *nonclassical depth* $\mathfrak{T}[\hat{\boldsymbol{\rho}}]$ of a CV state $\hat{\boldsymbol{\rho}}$ to quantify its nonclassicality:

$$\mathfrak{T}[\hat{\boldsymbol{\rho}}] := \frac{1 - s_m}{2} \tag{11}$$

where $s_m$ is the largest real number such that $W_s[\hat{\boldsymbol{\rho}}](X)$ is nonsingular and non-negative $\forall s < s_m$. In terms of $\mathfrak{T}[\hat{\boldsymbol{\rho}}]$, we may say that $\hat{\boldsymbol{\rho}}$ is nonclassical if $\mathfrak{T}[\hat{\boldsymbol{\rho}}] > 0$ and classical if

---

[2]In that case, it is immediate to check that $\chi[\hat{\rho}]$ is a Gaussian function too.

$\mathfrak{T}[\hat{\boldsymbol{\rho}}] = 0$. According to this definition, coherent states are the only classical pure states [40], while number states are highly nonclassical, with $|n\rangle$ having a P-function proportional to the $n$-th derivative of the delta distribution [27].

We should also mention that the Wigner function is often regarded as the closest approach to a classical description of a quantum state on phase space: it is always a nonsingular, normalized function, but for certain quantum states it may be not everywhere positive on phase space. For this reason, *Wigner negativity*, i.e. having a Wigner function that attains negative values in at least some regions of phase-space, can be encountered as an alternative definition of nonclassicality. However, Wigner negativity always implies P-nonclassicality, whereas a nonsingular, non-negative P-function guarantees a non-negative Wigner function. Furthermore, Gaussian states cannot exhibit Wigner negativity by definition, but Gaussian squeezed states are known to possess nonclassical features [41, 42]. We take these considerations as further reasons to back up our choice of nonclassicality (see also [43]).

Let us look in greater detail at the definition of P-nonclassicality for Gaussian states [44]. Since, for a Gaussian state $\hat{\boldsymbol{\rho}}$, by definition $\chi[\hat{\boldsymbol{\rho}}](\Lambda)$ is a Gaussian function on phase space, it is straightforward to conclude from Eq. (7) and Eq. (10) that $\hat{\boldsymbol{\rho}}$ is nonclassical if and only if the least eigenvalue $\lambda_-$ of its CM $\boldsymbol{\sigma}$ is smaller than $\frac{1}{2}$. In that case, the nonclassical depth of $\hat{\boldsymbol{\rho}}$ is given by:

$$\mathfrak{T}[\hat{\boldsymbol{\rho}}] = \frac{1}{2} - \lambda_- \tag{12}$$

Among classical Gaussian states we find all the (displaced) thermal states, including coherent states, while squeezed vacuum states are always nonclassical. In fact, squeezing is essentially the only source of P-nonclassicality in the Gaussian landscape.

## 3 Nonclassical steering with TMST states

### 3.1 Gaussian measurements and conditional states

Our first goal will be to describe the single-mode Gaussian states that can be prepared on Alice's mode, by Gaussian measurements on Bob's mode of a generic two-mode Gaussian state, and classical communication of the outcome.

A Gaussian measurement is implemented at the mathematical level by a positive operator-valued measure (POVM) $\{\hat{\boldsymbol{\Pi}}_\alpha\}_\alpha$ whose *effects* have Gaussian Wigner functions. In the single-mode case, we have:

$$\hat{\boldsymbol{\Pi}}_\alpha = \frac{1}{\pi} \hat{\mathbf{D}}(\alpha)\hat{\boldsymbol{\rho}}_M\hat{\mathbf{D}}^\dagger(\alpha), \tag{13}$$

where $\hat{\mathbf{D}}(\alpha) = e^{\alpha\hat{a}^\dagger - \alpha^*\hat{a}}$, $\alpha \in \mathbb{C}$, and $\hat{\boldsymbol{\rho}}_M$ is a single-mode Gaussian state with zero first-

moments vector and a CM $\boldsymbol{\sigma}_M$, that can always be written in the following form [3]:

$$\boldsymbol{\sigma}_M = \frac{1}{2\mu\mu_s} \begin{pmatrix} 1 + \kappa_s \cos\phi & -\kappa_s \sin\phi \\ -\kappa_s \sin\phi & 1 - \kappa_s \cos\phi \end{pmatrix} \tag{14a}$$

$$\mu = \mathrm{Tr}[\hat{\boldsymbol{\rho}}_M^2] \in [0,1] \tag{14b}$$

$$\mu_s = \frac{1}{1 + 2\sinh^2 r_m} \in [0,1] \tag{14c}$$

where $\mu$ is the purity of $\hat{\boldsymbol{\rho}}_M$, $\mu_s$ is the (single-mode) squeezing purity parameter and we introduced $\kappa_s = \sqrt{1 - \mu_s^2}$ for notational convenience. Here $r_m$ represents the single-mode squeezing parameter whereas $\phi \in [0, 2\pi)$ is the squeezing phase.

Let us now consider such a measurement performed by Bob on the second mode of a generic Gaussian state $\hat{\boldsymbol{\rho}}_{AB}$ of two modes. The probability of getting the outcome $\alpha$ is:

$$p_\alpha = \mathrm{Tr}_{AB}\left[\hat{\boldsymbol{\rho}}_{AB}\left(\mathbb{I}_A \otimes \hat{\boldsymbol{\Pi}}_\alpha\right)\right] \tag{15}$$

where $\mathbb{I}_A$ is the identity operator on the Hilbert space of Alice's mode. If now Bob communicates $\alpha$ to Alice, she can update the quantum state she uses to describe her mode, $\hat{\boldsymbol{\rho}}_A = \mathrm{Tr}_B[\hat{\boldsymbol{\rho}}_{AB}]$, according to:

$$\hat{\boldsymbol{\rho}}_A \to \hat{\boldsymbol{\rho}}_A^{(\alpha)} = \mathrm{Tr}_B[\hat{\boldsymbol{\rho}}_{AB}^{(\alpha)}] \tag{16a}$$

$$= \frac{1}{p_\alpha}\mathrm{Tr}_B\left[\hat{\boldsymbol{\rho}}_{AB}\left(\mathbb{I}_A \otimes \hat{\boldsymbol{\Pi}}_\alpha\right)\right] \tag{16b}$$

where $\hat{\boldsymbol{\rho}}_{AB}^{(\alpha)}$ is the quantum state of the two modes after the Gaussian measurement on the second mode resulted in the outcome $\alpha$. We will call $\hat{\boldsymbol{\rho}}_A^{(\alpha)}$ the *conditional state* of Alice's mode. Notice that, in order to specify $\hat{\boldsymbol{\rho}}_A^{(\alpha)}$, one doesn't need the decomposition of the Gaussian POVM into measurement operators, but it suffices to know the effects $\hat{\boldsymbol{\Pi}}_\alpha$ (unlike for the calculation of $\hat{\boldsymbol{\rho}}_{AB}^{(\alpha)}$).

The state $\hat{\boldsymbol{\rho}}_A^{(\alpha)}$ is still a Gaussian state [34, 45, 46], and if we write the CM $\boldsymbol{\sigma}$ of the initial state $\hat{\boldsymbol{\rho}}_{AB}$ in block form according to:

$$\boldsymbol{\sigma} = \begin{pmatrix} \mathbf{A} & \mathbf{C} \\ \mathbf{C}^T & \mathbf{B} \end{pmatrix} \tag{17}$$

then the conditional CM $\boldsymbol{\sigma}_A^{(\alpha)}$ of $\hat{\boldsymbol{\rho}}_A^{(\alpha)}$ is the Schur complement [47] of $\mathbf{B} + \boldsymbol{\sigma}_M$ in $\boldsymbol{\sigma}$:

$$\boldsymbol{\sigma}_A^{(\alpha)} = \boldsymbol{\sigma}/(\mathbf{B} + \boldsymbol{\sigma}_M) = \mathbf{A} - \mathbf{C}^T (\mathbf{B} + \boldsymbol{\sigma}_M)^{-1} \mathbf{C} \tag{18}$$

where $\boldsymbol{\sigma}_M$ is the CM of the POVM. The first-moments vector of the conditional state can also be calculated, but we do not need it because it doesn't influence the nonclassicality of $\hat{\boldsymbol{\rho}}_A^{(\alpha)}$.

---

[3]The option to choose such a parametrization is justified by the well-known result that any single-mode Gaussian state $\hat{\boldsymbol{\rho}}_M$ with zero first-moments vector is a single-mode squeezed thermal state.

The crucial point here is that the conditional CM $\boldsymbol{\sigma}_A^{(\alpha)}$ *does not depend on the outcome* $\alpha$ that Bob observed. This means that the nonclassical properties of the conditional state $\hat{\boldsymbol{\rho}}_A^{(\alpha)}$ of Alice's mode are completely specified by the CM $\boldsymbol{\sigma}$ of the initial state and the choice of Gaussian measurement made by Bob (which amounts to fixing $\boldsymbol{\sigma}_M$). However, one should keep in mind that Alice couldn't check that the CM of her mode after Bob's measurement is $\boldsymbol{\sigma}_A^{(\alpha)}$, unless he reports the observed value of $\alpha$ to her: in other words, she needs to know the updated first-moments vector of her mode. Since $\boldsymbol{\sigma}_A^{(\alpha)}$ doesn't really depend on $\alpha$, we shall rename it $\boldsymbol{\sigma}_A^c$ from now on, to distinguish it from the *unconditional* CM $\boldsymbol{\sigma}_A$ of Alice's state before the measurement, $\hat{\boldsymbol{\rho}}_A = \text{Tr}_B[\hat{\boldsymbol{\rho}}_{AB}]$. Since $\boldsymbol{\sigma}_A^c$ is a single-mode CM, it too can be recast in the form of Eq. (14a):

$$\boldsymbol{\sigma}_A^c = \frac{1}{2\mu_c\mu_{sc}} \begin{pmatrix} 1 + \kappa_{sc}\cos\phi_c & -\kappa_{sc}\sin\phi_c \\ -\kappa_{sc}\sin\phi_c & 1 - \kappa_{sc}\cos\phi_c \end{pmatrix} \tag{19}$$

with $\mu_c = \text{Tr}[(\hat{\boldsymbol{\rho}}_A^{(\alpha)})^2]$ and $\kappa_{sc} = \sqrt{1 - \mu_{sc}^2}$ as before.

We could also calculate the eigenvalues of $\boldsymbol{\sigma}_A^c$ from Eq. (19):

$$\lambda_\pm = \frac{1 \pm \kappa_{sc}}{2\mu_c\mu_{sc}} \tag{20}$$

Thus, recalling the the nonclassical depth (12), we can assert that a single-mode Gaussian state is nonclassical if and only if:

$$\lambda_- = \frac{1 - \kappa_{sc}}{2\mu_c\mu_{sc}} < \frac{1}{2} \quad \Longrightarrow \quad \mu_{sc} < \frac{2\mu_c}{1 + \mu_c^2} \tag{21}$$

Since $\mu_{sc} = (1 + \sinh^2 r_c)^{-1}$ where $r_c$ is the single-mode squeezing parameter, we can read Ineq. (21) as a lower bound on the single-mode squeezing, depending upon the purity of the state. Notice that the nonclassicality condition Ineq. (21) does not depend on the phase $\phi_c$.

## 3.2 General features of TMST states

Two-mode squeezed thermal states provide a simple, but sufficiently general and rich background to start our explorations [48–50]. They are described by a density operator:

$$\hat{\boldsymbol{\rho}}_{AB} := \hat{\mathcal{S}}^{(2)}(\xi) \left[ \hat{\boldsymbol{\nu}}_{th}(N_A) \otimes \hat{\boldsymbol{\nu}}_{th}(N_B) \right] \hat{\mathcal{S}}^{(2)}(\xi)^\dagger \tag{22}$$

where the two-mode squeezing unitary operator is defined as:

$$\hat{\mathcal{S}}^{(2)}(\xi) := e^{\xi\hat{a}^\dagger\hat{b}^\dagger - \xi^*\hat{a}\hat{b}} , \qquad \xi := re^{i\psi}$$

and $r$ is the two-mode squeezing parameter [4]. The single-mode thermal states $\hat{\boldsymbol{\nu}}_{th}(N)$, instead, are defined according to:

$$\hat{\boldsymbol{\nu}}_{th}(N) = \frac{1}{1+N} \sum_{n=0}^{\infty} \left( \frac{N}{1+N} \right)^n |n\rangle\langle n| \tag{23}$$

---

[4]In quantum optics, $r$ depends upon the power of the pumping beam, the interaction time and the second-order nonlinear electric susceptibility of the nonlinear optical element employed to prepare the state.

where $N$ the average number of photons. It is often practical to introduce *purity parameters*:

$$\mu(N) \; := \; \frac{1}{1 + 2N} \; , \qquad \mu_s(r) \; := \; \frac{1}{1 + 2\sinh^2 r} \tag{24}$$

With these definitions, $\mu(N)$ is precisely the purity of a single-mode thermal state $\hat{\boldsymbol{\nu}}_{th}(N)$, while $\mu_s(r)$ is a *two-mode squeezing purity parameter*, and it can be thought of as the purity of each mode in a two-mode squeezed vacuum state with squeezing parameter $r$: the larger $r$, the larger the entanglement, the smaller the purities of the partial traces. Note that they are a sufficiently vast class of two-mode Gaussian state to host both entangled *and* separable states.

Using the phase space formalism and the fact that unitary transformations generated by inhomogeneous quadratic hamiltonians act as affine symplectic transformations on the quadrature variables, one can deduce the form of the CM for a generic TMST state (we will assume a squeezing phase $\psi = 0$ from now on):

$$\boldsymbol{\sigma} \; = \; \begin{pmatrix} a & 0 & c & 0 \\ 0 & a & 0 & -c \\ c & 0 & b & 0 \\ 0 & -c & 0 & b \end{pmatrix} \tag{25}$$

$$\begin{cases} a & = & \dfrac{-\mu_A + \mu_B + (\mu_A + \mu_B)\cosh 2r}{4\mu_A\mu_B} \\[2mm] b & = & \dfrac{\mu_A - \mu_B + (\mu_A + \mu_B)\cosh 2r}{4\mu_A\mu_B} \\[2mm] c & = & \dfrac{(\mu_A + \mu_B)\sinh 2r}{4\mu_A\mu_B} \end{cases} \tag{26}$$

Another feature of TMST states that makes them handy in the study of remote generation of nonclassicality is the following. Consider the first mode of a TMST, controlled by Alice. Its quantum state is given by $\hat{\boldsymbol{\rho}}_A = \mathrm{Tr}_B[\hat{\boldsymbol{\rho}}_{AB}]$: this is still a Gaussian state, with CM proportional to the $2 \times 2$ identity matrix, $\boldsymbol{\sigma}_A = a \cdot \mathbb{I}_2$ and $a$ given by Eq. (26). Since $a \geq \frac{1}{2}$, according to our criterion for P-nonclassicality of Gaussian states, $\hat{\boldsymbol{\rho}}_A$ is always classical. The same holds true for $\hat{\boldsymbol{\rho}}_B$ of course, the reduced quantum state of the second mode, controlled by Bob. In other words, TMST states never possess any local nonclassicality.

## 3.3 Gaussian Steering triangoloids

Our next task is to determine analytically $\boldsymbol{\sigma}_A^c$ in the particular case of an initial TMST state $\hat{\boldsymbol{\rho}}_{AB}$. Thus we should determine the functional dependence of $\mu_c$, $\mu_{sc}$ and $\phi_c$ on the initial state's parameters $\mu_A, \mu_B, r$ and the POVM parameters $\mu, \mu_s, \phi$. According to Eq. (25), Eq. (26) and Eq. (18), for a TMST we have:

$$\boldsymbol{\sigma}_A^c \; = \; a \cdot \mathbb{I}_2 - c^2 \left[ \sigma_z \cdot (b \cdot \mathbb{I}_2 + \boldsymbol{\sigma}_M)^{-1} \cdot \sigma_z \right] \tag{27}$$

where $\sigma_z = \mathrm{diag}(1, -1)$. We now introduce two new parameters to clear up the formulae:

$$\alpha \; := \; b + \frac{1}{2\mu\mu_s} \; , \qquad \beta \; := \; \frac{\kappa_s}{2\mu\mu_s} \tag{28}$$

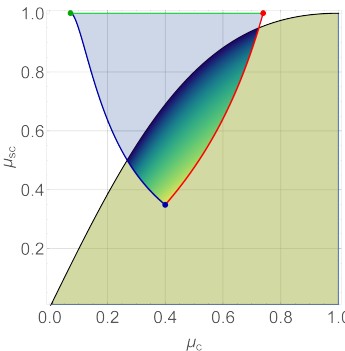 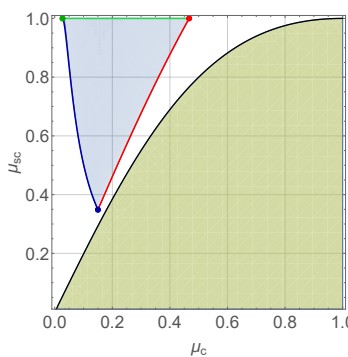 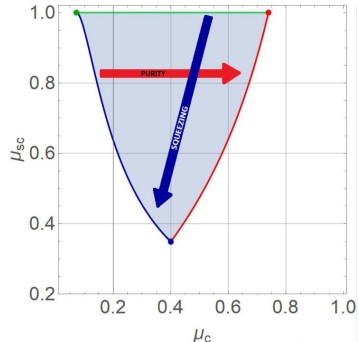

Figure 1: (Left): Triangoloid for TMST state with $\mu_A = \mu_B = 0.4$ and $r = 1.2$, $\mu_c$ is the purity of the conditional state, while $\mu_{sc} = (1 + 2\sinh^2 r_c)^{-1}$ quantifies squeezing of the conditional state. The light-brown region contains all nonclassical conditional states. (Middle): triangoloid for $\mu_A = \mu_B = 0.15$ and $r = 1.2$. (Right): triangoloid showing the directions of increasing purity of the measurement $\mu$ (red arrow) and increasing squeezing of the measurement $r_m$ (blue arrow).

with $\kappa_s = \sqrt{1 - \mu_s^2}$ as in Eq. (14a). Noting that $\alpha > \beta \geq 0$, we may write:

$$(b \cdot \mathbb{I}_2 + \boldsymbol{\sigma}_M)^{-1} = \frac{1}{\alpha^2 - \beta^2} \begin{pmatrix} \alpha + \beta\cos\phi & -\beta\sin\phi \\ -\beta\sin\phi & \alpha - \beta\cos\phi \end{pmatrix}$$

which can be inserted in Eq. (27) to arrive at:

$$\boldsymbol{\sigma}_A^c = a \cdot \mathbb{I}_2 - \frac{c^2}{\alpha^2 - \beta^2} \begin{pmatrix} \alpha - \beta\cos\phi & -\beta\sin\phi \\ -\beta\sin\phi & \alpha + \beta\cos\phi \end{pmatrix} \tag{29}$$

At this point, $\phi$ is still the phase of the measurement. However, note that $\mu_c$ and $\mu_{sc}$ can be retrieved from Eq. (19) using the following relations:

$$\det\left[\boldsymbol{\sigma}_A^c\right] = (2\mu_c)^{-2}, \qquad \mathrm{Tr}\left[\boldsymbol{\sigma}_A^c\right] = (\mu_c\mu_{sc})^{-1}. \tag{30}$$

We can use these relations to solve Eq. (29) for $\mu_c$ and $\mu_{sc}$:

$$\begin{aligned} \mu_c &= \frac{1}{2}\sqrt{\frac{\alpha^2 - \beta^2}{(c^2 - a\alpha)^2 - a^2\beta^2}} \\ \mu_{sc} &= \frac{\sqrt{(\alpha^2 - \beta^2)\left[(c^2 - a\alpha)^2 - a^2\beta^2\right]}}{a(\alpha^2 - \beta^2) - \alpha c^2} \end{aligned} \tag{31}$$

and we see that $\mu_c$ and $\mu_{sc}$ are independent of $\phi$, therefore the same holds true for the conditional nonclassicality, as implied by Ineq. (21). We also deduce that $\phi_c = \phi$, so that we can completely ignore the phase for TMST states.

We can display the results of Eq. (31) using *triangoloid plots*. For a given TMST state, i.e. for fixed values of $\mu_A$, $\mu_B$ and $r$, we plot the region in the parameters' space of $\boldsymbol{\sigma}_A^c$ containing all points $(\mu_c, \mu_{sc}) \in (0, 1] \times (0, 1]$ described by Eq. (31) for all possible Gaussian measurements on Bob's mode, or in other words, for all possible values of the parameters $\mu, \mu_s \in (0, 1]$ of the

measurement's CM. For $\mu_A = \mu_B = 0.4$ and $r = 1.2$, we obtain the triangular-shaped region in the left image of Fig. 1, delimited by red, green and blue sides. The light-brown region covering the bottom-right corner, instead, contains all values of $\mu_c$ and $\mu_{sc}$ corresponding to a nonclassical conditional state, according to Ineq. (21). Since the triangoloid intersects this nonclassical region, the chosen TMST state allows Bob to *steer* Alice's mode into a nonclassical state by means of some Gaussian measurements. In this case, the area of intersection is shaded according to the nonclassical depths of the conditional states, with lighter (yellow) colors corresponding to higher values of $\mathfrak{T}$.

On the other hand, the triangoloid associated with a TMST whose parameters are $\mu_A = \mu_B = 0.15$ and $r = 1.2$, does not intersect the nonclassical region as shown in the middle panel of Fig. 1: starting with this state, there is no Gaussian measurement that Bob can do on his mode to condition Alice's mode into a nonclassical state. We are therefore led to the following definition:

**Definition 1.** A TMST state is said to be *nonclassically steerable* from mode $B$ to mode $A$ if there exists a Gaussian measurement $\{\hat{\mathbf{\Pi}}_\alpha\}_{\alpha \in \mathbb{C}}$ on mode $B$ such that the conditional state $\hat{\rho}_A^{(\alpha)}$ of mode $A$ is nonclassical.

In order to gain some intuition about what kind of TMST states are nonclassically steerable, let us look closer at the triangoloid plots. We will list the relationships between the relevant points and sides of the triangoloid, and the corresponding measurements that achieve those conditional states:

- The red, rightmost side corresponds to $\mu = 1$ in the measurement's CM. Hence, these are conditional states of Alice's mode corresponding to (non orthogonal) projective measurements of Bob's mode on displaced single-mode squeezed vacuum states. The upper-right red vertex is attained for zero squeezing ($\mu_s = 1$), or in other words *heterodyne measurement*, implemented by projectors onto coherent states. Squeezing of the measurement increases towards the blue vertex ($\mu_s$ decreases). Note that this precisely correspond to an increasing of the conditional squeezing ($\mu_{sc}$ decreases too).

- The green, uppermost side corresponds to non squeezed POVMs ($\mu_s = 1$). Inserting this value in the second of Eq. (31), one can immediately deduce that $\mu_{sc} = 1$ for any TMST state: the conditional state has zero squeezing too, therefore it is always classical. The purity of the associated POVMs decreases ($\mu$ decreases) along this side from the red vertex to the green vertex, and correspondingly does the purity of the conditional state.

- The leftmost, blue side is not strictly part of the triangoloid, because it is attained only in some unphysical limit of the POVM. Specifically, if one renames the measurement's purity parameters as $\mu = tx$ and $\mu_s = x$, the parametric equation for the blue side as a function of the parameter $t \in \mathbb{R}^+$ is given by:

$$\lim_{x \to 0^+} \mu_c \left[ \mu_A, \mu_B, r; \mu = tx, \mu_s = x \right] \tag{32a}$$

$$\lim_{x \to 0^+} \mu_{sc} \left[ \mu_A, \mu_B, r; \mu = tx, \mu_s = x \right] \tag{32b}$$

The value of $t$ increases from $t = 0$ at the upper-left, green vertex, to $t \to +\infty$ towards the blue, bottom vertex. Note that the green vertex ($t = 0$) amounts to setting $\mu = 0$

before taking the limit: in this case, the conditional CM becomes independent of $\mu$, provided that $\mu \neq 0$. In such a limit, all POVM's effects $\hat{\boldsymbol{\Pi}}_\alpha$ approach the identity operator on mode $B$, which is equivalent to measuring without recording the outcome, hence we can describe the green vertex by the condition $\boldsymbol{\sigma}_A^c = \boldsymbol{\sigma}_A$.

- The blue, bottom vertex is the most important point for nonclassical steering. Indeed, one can infer graphically (and we will later prove it analytically) that this is the decisive point to establish whether the triangoloid intersects the nonclassical region or not. Formally, it amounts to taking the two limits $t \to +\infty$ and $x \to 0$ together in Eq. (31), so as to keep $\mu > 0$ and finite, and consequently $\mu_s \to 0^+$. However, it is simpler to describe it directly as the infinite measurement's squeezing limit ($\mu_s \to 0$) of Eq. (31); the conditional parameters $\mu_c$ and $\mu_{sc}$ become independent of $\mu \neq 0$ in this limit. Physically, this is achieved by projective measurements on the field quadratures (also known as *homodyne measurements*). Note that, since $\mu_c$ and $\mu_{sc}$ do not depend on the measurement's phase $\phi$, any choice of field quadrature of mode $B$ will lead to this point, but because $\phi_c = \phi$, different choices of $\phi$ yield distinct conditional states.

The trends we just listed are summarized by the right panel of Fig. 1: the red arrow shows the direction in which the conditional states in the triangoloid are associated with increasing purity of the Gaussian measurements that generated them, while the blue arrow indicates the direction of increasing squeezing of the associated Gaussian measurements. Relying on these qualitative considerations, we now prove the main result concerning nonclassical steering for TMST states:

**Proposition 1.** Given a generic TMST state $\hat{\boldsymbol{\rho}}_{AB}$, the nonclassicality of the conditional state $\hat{\boldsymbol{\rho}}_A^{(\alpha)}$ resulting from Gaussian measurement on mode $B$ is monotonically non-decreasing with the squeezing parameter $r_m$ of the measurement. In particular, among Gaussian measurements, any *field-quadrature projective measurement* is optimal to remotely generate nonclassicality with a TMST state and the TMST is nonclassically steerable from mode $B$ to mode $A$ if and only if its parameters fulfill the following inequality:

$$\varsigma_{A|B} > 1 \tag{33}$$

where we introduced the *nonclassical steerability* from $B$ to $A$:

$$\varsigma_{A|B} := \frac{\mu_A - \mu_B}{2} + \frac{\mu_A + \mu_B}{2} \cosh 2r . \tag{34}$$

*Proof.* Combining Eq. (31) and Eq. (20), one can express the nonclassicality of the conditional state, $\mathfrak{T} = \frac{1}{2} - \lambda_-$ where $\lambda_-$ is the smallest eigenvalue of $\boldsymbol{\sigma}_A^c$, as a function of $\mu_A, \mu_B, r, \mu$ and $\mu_s$. Explicit calculation of the derivative of $\mathfrak{T}$ with respect to $\mu_s = (1 + \sinh^2 r_m)^{-1}$ shows, by inspection, that it is always non-positive (under the assumptions $0 < \mu, \mu_s, \mu_A, \mu_B \leq 1$), therefore $\mathfrak{T}$ is monotonically non-decreasing with the measurement's squeezing $r_m$. Inequality (33) then follows from Ineq. (21) in the homodyne limit $\mu_s \to 0$ ($\mu \neq 0$). $\qquad\square$

## 3.4 Role of entanglement

An interesting question at this point is: does nonclassical steering imply (or is it implied by) entanglement? In order to answer this question, it is mandatory to recall that the Peres-Horodecki criterion, based on the negativity of the partially transposed quantum state, is a

necessary and sufficient condition for Gaussian entanglement [51]. Let us suppose that $\hat{\rho}_{AB}$ is a generic bipartite Gaussian state with CM $\sigma$. If we call:

$$\epsilon \ := \ \max\left[0, -\log(2\tilde{d}_-)\right] \tag{35}$$

the entanglement negativity, where $\tilde{d}_-$ is the smallest symplectic eigenvalue of the partially mirror-reflected $\sigma$, the condition for entanglement of $\hat{\rho}_{AB}$ is simply $\epsilon > 0$. In the case of a TMST state [52], one arrives at the following necessary and sufficient inequality for entanglement:

$$\frac{\sqrt{(\mu_A + \mu_B)^2 \cosh^2 2r - 4\mu_A\mu_B}}{2 - (\mu_A + \mu_B)\cosh 2r} > 1\,. \tag{36}$$

We can now provide the answer to the aforementioned question in the form of the following proposition:

**Proposition 2.** Given a generic TMST Gaussian state, entanglement is necessary, but not sufficient for it to be nonclassically steerable (in at least one direction).

*Proof.* We will show that $\varsigma_{A|B} > 1$ implies $\epsilon > 0$ and then we provide a counterexample to the inverse implication. Since initial entanglement is always a symmetric quantity in $\mu_A$ and $\mu_B$, we can assume $\varsigma \equiv \varsigma_{A|B} > 1$ so that mode $B$ can steer mode $A$ without any loss of generality. Let us re-express Ineq. (36) in terms of $\mu_A, \mu_B$ and $\varsigma$, by inverting the definition of the steering parameter:

$$\sqrt{(2\varsigma - \mu_A + \mu_B)^2 - 4\mu_A\mu_B} > 2 - (2\varsigma - \mu_A + \mu_B)\,. \tag{37}$$

From Ineq. (36) we already know that the left-hand side is real and non-negative. Then, if the right-hand side is strictly negative, $\epsilon > 0$ and we are done. Otherwise, suppose that $2\varsigma - \mu_A + \mu_B \leq 2$, so that the right-hand side of Ineq. (37) is also positive and we can square both sides and cancel some terms:

$$2\varsigma \ > \ 1 + \mu_A - \mu_B(1 - \mu_A)\,. \tag{38}$$

The right-hand side is clearly $\leq 2$, while the left-hand side of Ineq. (38) is always $> 2$ under the steerability hypothesis $\varsigma > 1$. This concludes the proof that $\varsigma > 1$ implies $\epsilon > 0$.

To show that the contrary is not necessarily true and entanglement is not sufficient for nonclassical steerability, let us limit ourselves to the symmetric case $\mu_A = \mu_B$, in which the entanglement condition Ineq. (36) reduces to:

$$\mu_A\sqrt{\cosh^2 2r - 1} \ > \ 1 - \mu_A\cosh 2r\,. \tag{39}$$

These states are *not* nonclassically steerable if and only if:

$$\varsigma_{A|B}[\mu_A, \mu_A, r] \ = \ \mu_A\cosh 2r \ < \ 1\,. \tag{40}$$

We can then solve jointly Ineq. (39) and Ineq. (40) by noting that the second allows us to square the first and then imposing again Ineq. (40) to finally arrive at:

$$\frac{1 + \mu_A^2}{2\mu_A} < \cosh 2r < \frac{1}{\mu_A}\,. \tag{41}$$

This constraint on $r$ admits solutions for any $0 < \mu_A \leq 1$, because the lower bound is always smaller than the upper bound in the allowed range of $\mu_A$, so *no matter the amount of two-mode squeezing, there exist infinitely many symmetric entangled initial states that cannot be used for nonclassical steering.* Asymmetric instances also exist: take for example $\mu_A = 0.5$ and $\mu_B = 0$. Then one can check that for $2 < \cosh 2r < 3$ both $\epsilon > 0$ and $\varsigma_{A|B} < 1$. Since $\mu_A > \mu_B$ in this case, $\varsigma_{B|A} < \varsigma_{A|B}$, so Nonclassical steering is forbidden also in the other direction. $\qquad\square$

### 3.5  Asymmetric nonclassical steering and further comments

Since Ineq. (33) is clearly asymmetric with respect to the two modes, it is possible to have TMST states that are nonclassically steerable just in one direction. For example, we can consider together the inequalities for nonclassical steerability from mode $B$ to $A$ and non-steerability from $A$ to $B$:

$$\mu_A - \mu_B + (\mu_A + \mu_B)\cosh 2r > 2\,,$$
$$\mu_B - \mu_A + (\mu_A + \mu_B)\cosh 2r < 2\,.$$

They can be re-expressed as:

$$\mu_A > \mu_B \ \wedge \ \frac{2 - \mu_A + \mu_B}{\mu_A + \mu_B} < \cosh 2r < \frac{2 + \mu_A - \mu_B}{\mu_A + \mu_B}\,. \tag{42}$$

This can happen for arbitrarily large values of the two-mode squeezing parameter $r$, since it suffices to choose $\mu_1 = \frac{3}{2n}$ and $\mu_2 = \frac{1}{2n}$ with $n$ a large natural number, to have $n - \frac{1}{2} < \cosh 2r < n + \frac{1}{2}$. At fixed values of $\mu_A$ and $\mu_B$, instead, we see that there is a minimum value of $r$ after which nonclassical steering becomes possible, but in only one direction (mode $B$ can steer mode $A$ if $\mu_A > \mu_B$ and vice versa otherwise), until a maximum value of $r$ is exceeded and then the ability of nonclassical steering becomes necessarily symmetric for all larger values of $r$.

We can also recast the condition $\varsigma_{A|B} > 1$ in terms of the mean number of squeezing photons per mode, $N_s = \sinh^2 r$, and the mean number of thermal photons in each mode, $N_A = \frac{1 - \mu_A}{2\mu_A}$ (and similarly for $N_B$):

$$N_s \ > \ \frac{N_A (1 + 2N_B)}{1 + N_A + N_B}\,. \tag{43}$$

We can deduce some useful characterization exploiting the above inequality:

- If $N_A = 0$, i.e. the mode to be steered has zero thermal noise, then any amount of initial squeezing ($N_s > 0$) is enough to ensure nonclassical steerability from $B$ to $A$. Graphically, $N_A = 0$ corresponds to triangoloids whose red, upper-right vertex is in $(\mu_c, \mu_{sc}) = (1, 1)$, so that graphically it is clear that they always intersect the nonclassical region.

- If $N_B = 0$, then the mode to be measured has zero thermal noise and:

  $$N_s > \frac{N_A}{1 + N_A}$$

  In particular, $N_s > 1$, or $r > \operatorname{asinh}(1) \simeq 0.8814$ guarantees nonclassical steerability for any $N_A$.

- If $N_A \to +\infty$ the condition simplifies to:

$$N_s \; > \; 2N_B + 1$$

while, for $N_B \to +\infty$:

$$N_s \; > \; 2N_A$$

- For symmetric TMST states, i.e. for $N_A = N_B$, the condition simplifies to $N_s > N_A$: the number of nonclassical resources per mode ($N_s$) should be strictly larger than the number of thermal photons per mode.

## 4 Application to noisy propagation of TWB states

We will now discuss a partially realistic scenario to test the notion of nonclassical steering. It would involve Bob preparing a correlated two-mode state, sending one of the modes to Alice through an inevitably noisy channel, and then trying to nonclassically steer her mode at a distance by Gaussian measurement on the mode he kept. Clearly, then, the residual noise acting directly on Bob's mode can be neglected: if it is detrimental, he can perform the measurement on mode $B$ just after the preparation, before any noise can spoil his state, and then send the conditional state of mode $A$ to Alice. The noise acting on mode $A$, instead, is truly inescapable, as always when one tries to broadcast quantum states. If we call $\hat{\boldsymbol{\rho}}_{AB}$ the generic bipartite state of the two modes and $\mathcal{E}_A(t)$ the noise map acting on mode $A$, then the state of the two modes after a propagation time $t$ of mode $A$ is:

$$\hat{\boldsymbol{\rho}}_{AB}(t) \;\; = \;\; (\mathcal{E}_A \otimes \mathbb{I}_B)\left[\hat{\boldsymbol{\rho}}_{AB}(0)\right] \tag{44}$$

with $\hat{\boldsymbol{\rho}}_{AB}(0) = \hat{\boldsymbol{\rho}}_{AB}$. Bob can perform a measurement described by the POVM $\{\hat{\boldsymbol{\Pi}}_\alpha\}_{\alpha \in \mathbb{C}}$ (not necessarily Gaussian at this stage) on his mode either at time $t = 0$, just after the preparation, or at a later time $t$. Using a Kraus decomposition of $\mathcal{E}_A$, it is simple to show that the conditional state that will arrive to Alice, $\hat{\boldsymbol{\rho}}_A^{(\alpha)}(t)$, is the same in both cases: he doesn't gain anything by waiting, but he can delay his choice of measurement without loosing any power in his task of preparing a nonclassical state at Alice's place.

Let us now discuss the effects of such a noisy propagation on triangoloids and on the nonclassical steerability condition, Ineq. (33). We will assume that the quantum state of the two modes immediately after its preparation is a TMST state with zero thermal noise on both modes, i.e. it is a twin-beam state (TWB, also known as two-mode squeezed vacuum state):

$$|r\rangle\rangle \;\; = \;\; e^{r(\hat{a}^\dagger \hat{b}^\dagger - \hat{a}\hat{b})} \, |0\rangle_A \otimes |0\rangle_B \tag{45}$$

with $r \in \mathbb{R}^+$. The TWB states are maximally entangled states of two modes at fixed energy. Indeed, they can be written in the number eigenbasis of the two modes as:

$$|r\rangle\rangle \;\; = \;\; \sqrt{1 - \lambda^2} \sum_{n=0}^{\infty} \lambda^n \, |n\rangle \otimes |n\rangle \tag{46}$$

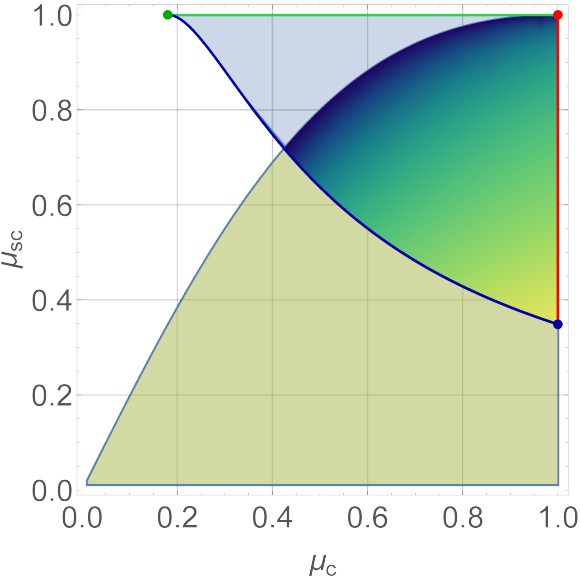

Figure 2: Triangoloid for TWB state, with $\mu_A = \mu_B = 1$ and $r = 1.2$.

where $\lambda = \tanh r$, hence they manifest perfect correlations in photon-counting measurements and they provide one of the few ways to generate higher photon number states. Their entanglement negativity is $\epsilon_{\text{TWB}}[r] = 2r$, hence all TWB states are entangled, as long as $r > 0$. Moreover, they are always nonclassically steerable because they have $N_A = N_B = 0$. Their triangoloids have a right-angled red vertex, in $\mu_c = \mu_{sc} = 1$, as in Fig. 2.

The noisy propagation of mode $A$ of the TWB state inside an optical medium can be modeled by a coupling of the mode with a non-zero temperature reservoir, i.e. a bath of infinitely many, decoupled oscillators thermalized at the same temperature [53]. The dynamics can be described in terms of the Master equation, also known as *Lindblad Equation*, which for an $n$-mode state $\hat{\boldsymbol{\rho}}$ reads:

$$\frac{\mathrm{d}\hat{\boldsymbol{\rho}}(t)}{\mathrm{d}t} = \sum_{k=1}^{n} \frac{\Gamma_k}{2} \left\{ (N_{th,k} + 1)\mathcal{L}[\hat{a}_k] + N_{th,k}\mathcal{L}[\hat{a}_k^\dagger] \right\} \hat{\boldsymbol{\rho}}(t) \tag{47}$$

where $\Gamma_k \geq 0$ is the damping rate for the $k$-th mode, taking into account the couplings between the bath and the mode, $N_{th,k} \in \mathbb{R}^+$ is the mean photon-number density per unit frequency around the frequency of mode $k$ interacting with the bath, and $\mathcal{L}$ is the *Lindblad superoperator*:

$$\mathcal{L}[\hat{O}]\hat{\boldsymbol{\rho}} = 2\hat{O}\hat{\boldsymbol{\rho}}\hat{O}^\dagger - \hat{O}^\dagger\hat{O}\hat{\boldsymbol{\rho}} - \hat{\boldsymbol{\rho}}\hat{O}^\dagger\hat{O} \tag{48}$$

Passing to the phase space formalism through a differential representation of the mode operators in Eq. (47), one can derive a Fokker-Planck equation [54] for the Wigner function of $\hat{\boldsymbol{\rho}}(t)$. When the initial state is Gaussian, it will stay Gaussian throughout the evolution and a simple solution can be derived for the time evolution $\boldsymbol{\sigma}_t$ of the CM of $\hat{\boldsymbol{\rho}}(t)$:

$$\boldsymbol{\sigma}_t = \mathbb{G}_t^{1/2}\boldsymbol{\sigma}_0\mathbb{G}_t^{1/2} + (\mathbb{I}_{2n} - \mathbb{G}_t)\boldsymbol{\sigma}_\infty \tag{49a}$$

$$\mathbb{G}_t \; := \; \bigoplus_{k=1}^{n} e^{-\Gamma_k t} \mathbb{I}_2 \tag{49b}$$

$$\boldsymbol{\sigma}_\infty \; := \; \bigoplus_{k=1}^{n} \left( N_k + \frac{1}{2} \right) \mathbb{I}_2 \tag{49c}$$

In Eq. (49a), $\boldsymbol{\sigma}_0$ is the initial CM, while $\boldsymbol{\sigma}_t$ is the CM after a propagation time $t$ and $\boldsymbol{\sigma}_\infty$ is the asymptotic CM, corresponding to complete thermalization of each mode of the state with the corresponding bath of oscillators.

In our case, we shall assume that only mode $A$ interacts with a bath, therefore $\Gamma_B = 0$ and we rename $\Gamma_A = \Gamma$. We can also call $N_{th}$ the average number density of thermal photons in the bath at the frequency of mode $A$. Moreover, the initial CM $\boldsymbol{\sigma}_0$ is the CM of a TWB state, which is in canonical form with parameters:

$$
\begin{aligned}
a^0 \; &= \; b^0 \; = \; N_s + \frac{1}{2} \\
c_1^0 \; &= \; -c_2^0 \; = \; \sqrt{N_s(1 + N_s)}
\end{aligned}
\tag{50}
$$

and $N_s = \sinh^2 r$ as usual. Inserting the corresponding $\boldsymbol{\sigma}_0$, $\mathbb{G}_t = \left( e^{-\Gamma t} \mathbb{I}_2 \right) \oplus \mathbb{I}_2$ and $\boldsymbol{\sigma}_\infty = \left( N_{th} + \frac{1}{2} \right) \mathbb{I}_4$ in Eq. (49a), we find the CM of the two modes at time $t$:

$$
\boldsymbol{\sigma}_t \; = \;
\begin{pmatrix}
a' & 0 & c' & 0 \\
0 & a' & 0 & -c' \\
c' & 0 & b' & 0 \\
0 & -c' & 0 & b'
\end{pmatrix}
\tag{51}
$$

with time-dependent parameters $a'$, $b'$ and $c'$ given by:

$$a'(t) = N_{th} + \frac{1}{2} + e^{-\Gamma t}(N_s - N_{th}), \tag{52a}$$

$$b' = N_s + \frac{1}{2}, \tag{52b}$$

$$c'(t) = \sqrt{e^{-\Gamma t} N_s(1 + N_s)}. \tag{52c}$$

The initial TWB state, after propagation of mode $A$ for a time $t$ in the thermal environment, has become a generic TMST state, with a CM $\boldsymbol{\sigma}_t$ in canonical form with $c'_1 = -c'_2 = c'$. We can now compare Eq. (26) with Eq. (52) to get the new purity parameters of the two modes, $\mu'_A$ and $\mu'_B$, and the new two-mode squeezing parameter $r'$. The result is:

$$\mu'_A(t) \; = \; \frac{e^{\Gamma t}}{(N_s - N_{th})(1 - e^{\Gamma t}) + \sqrt{[N_s - N_{th} + e^{\Gamma t}(1 + N_s + N_{th})]^2 - 4e^{\Gamma t} N_s(1 + N_s)}}, \tag{53a}$$

$$\mu'_B(t) \; = \; \frac{e^{\Gamma t}}{(N_{th} - N_s)(1 - e^{\Gamma t}) + \sqrt{[N_s - N_{th} + e^{\Gamma t}(1 + N_s + N_{th})]^2 - 4e^{\Gamma t} N_s(1 + N_s)}}, \tag{53b}$$

$$r'(t) \;=\; \frac{1}{2} \, \mathrm{arccosh} \left[ \frac{N_s - N_{th} + e^{\Gamma t}(1 + N_s + N_{th})}{\sqrt{[N_s - N_{th} + e^{\Gamma t}(1 + N_s + N_{th})]^2 - 4e^{\Gamma t} N_s(1 + N_s)}} \right]. \tag{53c}$$

All these quantities decrease monotonically with propagation time $t$. While $r'(t)$ drops to 0, implying that the state asymptotically becomes factorized, $\mu'_A(t)$ and $\mu'_B(t)$ approach asymptotic values given by:

$$\lim_{t \to +\infty} \mu'_A(t) \;=\; \frac{1}{1 + 2N_{th}}, \tag{54a}$$

$$\lim_{t \to +\infty} \mu'_B(t) \;=\; \frac{1}{1 + 2N_s}. \tag{54b}$$

We can put to use Ineq. (33) to decide whether the state after propagation of mode $A$ for a time $t$ is still nonclassically steerable or not. Computing the nonclassical steerability $\varsigma_{A|B}$ from Eq. (34) and Eq. (53), we find the *maximum propagation time for nonclassical steering*, $t_{\mathrm{ns}}$, after which Bob can no longer steer Alice's mode into a nonclassical state:

$$t_{\mathrm{ns}} \;=\; \frac{1}{\Gamma} \, \log \left[ 1 + \frac{N_s}{N_{th}(1 + 2N_s)} \right]. \tag{55}$$

In general, $t_{\mathrm{ns}}$ is smaller than the maximum time $t_{\mathrm{ent}}$ after which the modes are no longer entangled, which was computed for example in [55] for the case of a TWB having both modes interacting with reservoires at the same temperature and with equal damping rates. We explicitly calculated $t_{\mathrm{ent}}$ for our situation using entanglement negativity:

$$t_{\mathrm{ent}} \;=\; \frac{1}{\Gamma} \, \log \left( 1 + \frac{1}{N_{th}} \right). \tag{56}$$

Somehow surprisingly, it does not depend on $N_s$ and it is always greater than the upper bound on $t_{\mathrm{ns}}$, even in the limit of infinite initial entanglement, $N_s \to +\infty$. A similar result has been obtained in [56] (in particular, see eq. (5.3) therein and the subsequent discussion). Here we stress the fact that there is always a non-zero time lapse, between $t_{\mathrm{ent}}$ and $t_{\mathrm{ns}}$, during which we observe entangled TMST states that are *not* nonclassically steerable: this is in perfect agreement with our result, Proposition 2, reinforcing the idea that being nonclassically steerable is a stronger condition than entanglement for TMST states, and shows that such states are not rare and odd exceptions, but they arise quite naturally.

We can exploit the triangoloid plots to monitor the evolution of the set of conditional states that can be prepared on mode $A$ at any given time (or, equivalently, the evolution of the set of conditional states generated just after the preparation of the TWB). In Fig. 3 we depicted a time sequence of triangoloids arising from an initial TWB state with $N_s = 1$, for a damping rate $\Gamma = 0.1$ and $N_{th} = 0.2$. They shrink progressively, until they completely get out of the nonclassical region at $t = t_{\mathrm{ns}}$. At later times, they continue to contract towards a point on the upper, green side; indeed, for $t \to +\infty$, the two-mode state becomes factorized and the state of mode $A$ cannot be conditioned by measurements on mode $B$, being just a thermal state with purity given by Eq. (54a). Note that the impression that they contract without drifting is not true in general, but only for some choices of $N_s$ and $N_{th}$.

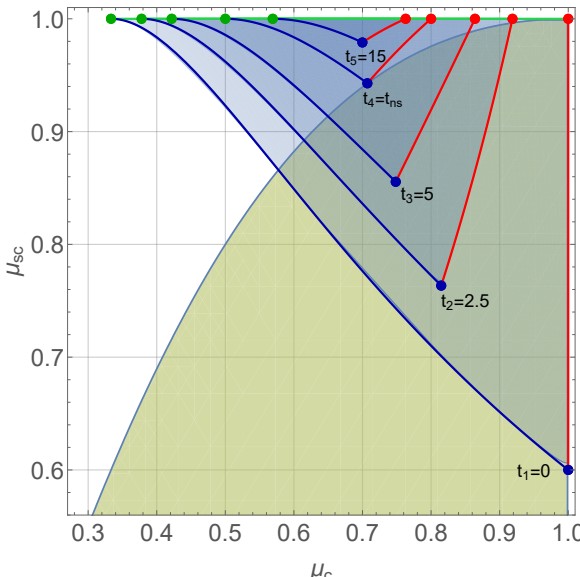

Figure 3: Time sequence of triangoloids, starting from a TWB state with $N_s = \sinh^2 r = 1$. The damping rate of the noisy channel acting on mode $A$ is $\Gamma = 0.1$ and the average number density of thermal photons is $N_{th} = 0.2$. At $t = t_{\rm ns}$ and for all greater times, the overlap with the nonclassical region (light-brown) vanishes.

## 5 The notion of nonclassical steering for a generic two-mode Gaussian state

The main conceptual difficulty we may encounter in generalizing the idea of nonclassical steering to all two-mode Gaussian state arises from single-mode squeezing: for a general Gaussian state $\hat{\boldsymbol{\rho}}_{AB}$ of two modes, the *unconditional* quantum state $\hat{\boldsymbol{\rho}}_A = \text{Tr}_B[\hat{\boldsymbol{\rho}}_{AB}]$ of mode $A$ may already be nonclassical due to single-mode squeezing. However, as we are trying to capture a type of *quantum correlations*, we should be able to perform local unitary operations without affecting them. In our context, we may freely perform Local Gaussian Unitary Transformations (LGUTs) on the two modes in order to bring $\hat{\boldsymbol{\rho}}_{AB}$ into a simpler form. In particular it is well understood that, by means of LGUTs, any two-mode Gaussian state can always be brought into the so-called *canonical form* [34, 57, 58], for which the generic CM $\boldsymbol{\sigma}$, written in block form as in Eq. (17), has:

$$\mathbf{A} = a \cdot \mathbb{I}_2, \quad \mathbf{B} = b \cdot \mathbb{I}_2, \quad \mathbf{C} = \text{diag}(c_1, c_2). \tag{57}$$

Here $a, b, c_1, c_2 \in \mathbb{R}$ are truly independent real parameters, but they nevertheless have to obey the constraints required by the positivity of $\boldsymbol{\sigma}$ and by Ineq. (9). The UR imply that $a, b \geq \frac{1}{2}$, hence the unconditional states $\hat{\boldsymbol{\rho}}_A = \text{Tr}_B[\hat{\boldsymbol{\rho}}_{AB}]$ and $\hat{\boldsymbol{\rho}}_B = \text{Tr}_A[\hat{\boldsymbol{\rho}}_{AB}]$ of both modes are still classical, for any two-mode Gaussian state $\hat{\boldsymbol{\rho}}_{AB}$ in canonical form. This observation let us suggest the following generalization of Def. 1:

**Definition 2.** A two-mode Gaussian state $\hat{\boldsymbol{\rho}}_{AB}$ in canonical form is called *weakly nonclassically steerable* (WNS) from mode $B$ to mode $A$ ($B \to A$) if there exists a Gaussian positive operator-valued measure (POVM) $\{\hat{\boldsymbol{\Pi}}_\alpha\}_{\alpha \in \mathbb{C}}$ on mode $B$ such that the *conditional state of mode $A$*, $\hat{\boldsymbol{\rho}}_A^{(\alpha)}$, is a *nonclassical state*.

We can also directly generalize Proposition 1:

**Proposition 3.** The least classical conditional state $\hat{\rho}_A^{(\alpha)}$ of mode $A$ resulting from conditioning upon Gaussian measurements on mode $B$ of a two-mode Gaussian state $\hat{\rho}_{AB}$ in canonical form is generated by a field-quadrature measurement on mode $B$, either of the $\hat{x}_B$ quadrature if $|c_2| \geq |c_1|$, or of the $\hat{p}_B$ quadrature otherwise. In particular, $\hat{\rho}_{AB}$ is WNS ($B \to A$) if and only if the parameters of its CM satisfy:

$$a - \frac{c^2}{b} \; < \; \frac{1}{2}, \quad c = \max\{|c_1|, |c_2|\}. \tag{58}$$

*Proof.* Inserting equations (57) for the canonical form in the general formula (18) for the conditional CM $\boldsymbol{\sigma}_A^c$, we observe that, since $\mathbf{A}$ is diagonal, the smallest eigenvalue of $\boldsymbol{\sigma}_A^c$ is minimized (over all possible CMs $\boldsymbol{\sigma}_M$) when the greatest eigenvalue $\lambda_M$ of $\mathbf{C}^T(\mathbf{B} + \boldsymbol{\sigma}_M)^{-1}\mathbf{C}$ attains its supremum, which is positive semidefinite, hence $\lambda_M \geq 0$. By explicit calculation, to maximize $\lambda_M$ the measurement's phase has to be $\phi = 0$ if $|c_2| \geq |c_1|$, and $\phi = \pi$ otherwise. Once the phase is settled to one of these values, $\lambda_M$ is a monotonic decreasing function of $\mu_s$, as can be checked by inspection of its first derivative with respect to $\mu_s$. Therefore, $\lambda_M$ is further maximized in the limit $\mu_s \to 0$, for which the value of $\mu(\neq 0)$ becomes irrelevant and the Gaussian POVM $\hat{\boldsymbol{\Pi}}_\alpha$ reduces to the spectral measure of the $\hat{x}$ quadrature for $\phi = 0$, and of the $\hat{p}$ quadrature for $\phi = \pi$. As for Eq. 58, note that if $c = |c_2| \geq |c_1|$, then we can fix the POVM's phase to $\phi = 0$ and, for $\mu_s \to 0$, we explicitly work out the minimum of the smallest eigenvalue of $\boldsymbol{\sigma}_A^c$:

$$\min\{\lambda_m\} \;=\; a - \frac{c^2}{b}.$$

This has to fulfill $\min\{\lambda_m\} < 1/2$ in order for the state $\hat{\rho}_{AB}$ to be nonclassically steerable, as stated by Eq. (58). Otherwise, if $c = |c_1| > |c_2|$, we choose $\phi = \pi$ to arrive at the same conclusion. $\qquad\square$

In switching from TMST states to generic states in canonical form, we called *weak* this generalized notion of nonclassical steering. The reason is that it does not imply entanglement, as we showed with some examples of parameters and also with explicit constructions of separable states that are nevertheless WNS (see Appendix A.1). A question may arise now on whether WNS is related to a more general class of quantum correlations, such as Gaussian Quantum Discord (GQD) [59–63]. We remark that Gaussian states with zero GQD, being factorized, are obviously *not* WNS. A reasonable guess, however, could be to expect a strictly positive lower bound to GQD for states exhibiting WNS. By construction of explicit counterexamples, we showed that this is not the case (see Appendix A.2).

Motivated by these findings, we shall introduce a tighter notion of nonclassical steering:

**Proposition 4.** A two-mode Gaussian state $\hat{\rho}_{AB}$ in canonical form is called *strongly nonclassically steerable* (SNS) from mode $B$ to mode $A$ if *any* field-quadrature measurement on mode $B$ generates a nonclassical conditional state of mode $A$. A necessary and sufficient condition for $\hat{\rho}_{AB}$ to be SNS is:

$$a - \frac{c'^2}{b} \; < \; \frac{1}{2}, \quad c' = \min\{|c_1|, |c_2|\} \tag{59}$$

where $a, b, c_1, c_2$ are the parameters of its CM in canonical form.

*Proof.* We follow the proof of Proposition 3. Among all quadrature measurements on mode $B$, the one leading to the least nonclassical conditional state of mode $A$ corresponds to the "wrong" choice of measurement's phase ($\phi = \pi$ for $|c_2| \geq |c_1|$ and $\phi = 0$ otherwise). Therefore, it suffices to require the smallest eigenvalue of $\boldsymbol{\sigma}_A^c$ to be smaller than $\frac{1}{2}$ also in this case, which gives Ineq. (59). $\qquad\qquad\qquad\qquad\qquad\qquad\qquad\qquad\qquad\qquad\qquad\qquad\qquad\qquad\qquad\square$

Comparing Ineq. (58) and Ineq. (59), we immediately conclude that weak and strong non-classical steering coincide precisely for the class of TMST states, since they are all and only those states in canonical form with $c_1 = -c_2$.

We now seek a generalization of weak and strong nonclassical steering to *all* Gaussian states of two modes. We recall once again that any two-mode Gaussian state can be brought to its *unique* canonical form through LGUTs without altering the correlations, thus we can extend the definitions in the following way:

**Definition 3.** A generic two-mode Gaussian state $\hat{\boldsymbol{\rho}}_{AB}$ is called weakly (strongly) nonclassically steerable if the *unique* Gaussian state $\hat{\boldsymbol{\rho}}'_{AB}$ *in canonical form* related to $\hat{\boldsymbol{\rho}}_{AB}$ by LGUTs is weakly (strongly) nonclassically steerable.

As for the results regarding the necessary and sufficient conditions for WNS/SNS, we have to define the effect of LGUTs on $\boldsymbol{\sigma}_A^c$. Given that any Gaussian unitary transformation is implemented on phase space by a symplectic linear transformation and vice versa, a LGUT on a two-mode system is described by a direct sum $S_A \oplus S_B$ of $2 \times 2$ matrices acting on quantum phase space, where $S_{A(B)} \in \mathrm{SL}_{A(B)}(2)$. The $2 \times 2$ blocks of a generic CM $\boldsymbol{\sigma}$, written as in Eq. (17) transform according to:

$$\mathbf{A}' = S_A \mathbf{A} S_A^T, \quad \mathbf{B}' = S_B \mathbf{A} S_B^T, \quad \mathbf{C}' = S_A \mathbf{C} S_B^T. \tag{60}$$

If $S_A \oplus S_B$ brings the initial $\boldsymbol{\sigma}$ in canonical form, then we have $\mathbf{A}' = a' \cdot \mathbb{I}_2$, $\mathbf{B}' = b' \cdot \mathbb{I}_2$ and $\mathbf{C}' = \mathrm{diag}(c'_1, c'_2)$. We can rearrange the conditional CM $\boldsymbol{\sigma}_A^c$ resulting from a Gaussian measurement with CM $\boldsymbol{\sigma}_M$ on the initial state with CM $\boldsymbol{\sigma}$ as:

$$\boldsymbol{\sigma}_A^c = S_A^T \left[ \mathbf{A}' - \mathbf{C}' \left( \mathbf{B}' + \boldsymbol{\sigma}'_M \right)^{-1} \mathbf{C}'^T \right] S_A \tag{61}$$

where we redefined the CM of the measurement as $\boldsymbol{\sigma}'_M = S_B^T \boldsymbol{\sigma}_M S_B$. We deduce that, for what concerns the conditional state of mode $A$, the measurement associated with $\boldsymbol{\sigma}_M$ acts on the two-mode state with CM $\boldsymbol{\sigma}$ in the same way as the measurement $\boldsymbol{\sigma}'_M$ acts on the canonical form state related to $\boldsymbol{\sigma}$, followed by a transformation induced by $S_A$ on the resulting conditional CM. Hence, the action of $S_A$ does not interfere with the steering process and we can simply factor it out. At the same time, as long as $S_B$ doesn't involve infinite squeezing, we can still reproduce the desired limit of $\boldsymbol{\sigma}'_M$, acting on the state in canonical form, by an infinite squeezing limit of $\boldsymbol{\sigma}_M$ with a suitable phase. We can finally replace $a, b, c_1, c_2$ in Ineq. (58) and Ineq. (59) with their expressions in terms of symplectic invariants [57] to arrive at the most general form of the necessary and sufficient conditions for WNS and SNS:

$$
\begin{aligned}
I_1 &= a^2 \,, \quad I_2 = b^2 \,, \\
I_3 &= c_1 c_2 \,, \quad I_4 = (ab - c_1{}^2)(ab - c_2{}^2) \,.
\end{aligned}
\tag{62}
$$

Indeed, these are the only independent combinations of the canonical parameters $a, b, c_1, c_2$ that are invariant under all LGUTs.

**Proposition 5.** Given any two-mode Gaussian state $\hat{\boldsymbol{\rho}}_{AB}$, it is WNS from mode $B \to A$ if and only if its symplectic invariants satisfy the inequality:

$$\frac{I' - \sqrt{I'^2 - 4I_1 I_2 I_4}}{2 I_2 \sqrt{I_1}} < \frac{1}{2} \tag{63}$$

while it is SNS from mode $B \to A$ if and only if they fulfill:

$$\frac{I' + \sqrt{I'^2 - 4I_1 I_2 I_4}}{2 I_2 \sqrt{I_1}} < \frac{1}{2} \tag{64}$$

where $I' = I_1 I_2 - I_3^2 + I_4$.

Clearly, strong nonclassical steering implies weak nonclassical steering. It deserves its name because it also implies entanglement, but we will prove this indirectly, via a stronger result:

**Theorem 1.** *Any two-mode Gaussian state $\hat{\boldsymbol{\rho}}_{AB}$ that is SNS from mode $B$ to mode $A$ is also EPR-steerable in the same direction. In particular, it must be entangled.*

*Proof.* Following [11], a two-mode Gaussian state is EPR-steerable from mode $B$ to mode $A$ by Gaussian measurements if and only if its CM *violates* the inequality:

$$\boldsymbol{\sigma} + \frac{i}{2} \boldsymbol{\omega}_A \oplus \mathbb{0}_B \geq 0 \tag{65}$$

where $\boldsymbol{\sigma}$ is the CM of $\hat{\boldsymbol{\rho}}_{AB}$ and $\mathbb{0}_B$ is the zero matrix on phase space of mode $B$. Exploiting LGUT-invariance, we can restrict the comparison between EPR-steerability and SNS to Gaussian states in canonical form. In this case, keeping in mind that $a > \frac{1}{2}$, violation of the above inequality reduces to [11,64]:

$$\left(a - \frac{c_1^2}{b}\right)\left(a - \frac{c_2^2}{b}\right) < \frac{1}{4} \tag{66}$$

which is certainly true under the SNS Ineq. (59). $\qquad\square$

One might wonder whether the EPR-steerability condition Ineq. (66) can be fulfilled by *physical* parameters $a, b, c_1, c_2$ that nevertheless violate the SNS condition Ineq. (59): in other words, if there actually exist two-mode Gaussian states which are EPR-steerable but not strongly nonclassically steerable in the same direction. We confirmed that this is the case with an explicit example, provided in Appendix A.3. It is also clear, from the proof of Theorem 1 we just presented, that TMST states are EPR-steerable from one mode to the other *if and only if* they are nonclassically steerable in the same direction, so that the three notions of steering coincide for them and Ineq. (66) for EPR steering takes the simpler form of Ineq. (33) that we derived for nonclassical steering. This observation suggests a new, surprising role for the notion of P-nonclassicality: Alice can be certain that the initially shared TMST state was indeed entangled if and only if the conditional state of her mode is nonclassical; we foresee that this fact could find applications in one-sided device-independent quantum key distribution [12], especially in light of the rich variety of techniques developed to detect nonclassicality (see [65] and references therein). Moreover, in light of this observation for TMST states, Proposition 2 amounts to the well-known fact that EPR-steerability is generally a

stronger requirement than entanglement. However, the proof we provided adds quantitative aspects to these considerations: for example, it shows that, for unconstrained local purities, there is no lower bound on two-mode squeezing that guarantees an entangled TMST state to be EPR-steerable, as one can also see from Ineq. (41).

As a further comment on the asymmetry of nonclassical steering, note that the WNS/SNS Ineq. (63) from $B$ to $A$ is tighter than the corresponding inequality from $A$ to $B$ (corresponding to exchanging $I_1$ and $I_2$ in Ineq. (63)) if and only if $I_1 > I_2$. But $I_1$ and $I_2$ are inversely proportional to the squares of the purities of the partial traces [5]:

$$I_{A(B)} \;=\; \frac{1}{4\mu_{1(2)}^2} \;, \qquad \mu_{1(2)} \;=\; \mathrm{Tr}[\hat{\boldsymbol{\rho}}_{A(B)}^2] \tag{67}$$

where $\hat{\boldsymbol{\rho}}_A = \mathrm{Tr}_B[\hat{\boldsymbol{\rho}}_{AB}]$ and $\hat{\boldsymbol{\rho}}_B = \mathrm{Tr}_A[\hat{\boldsymbol{\rho}}_{AB}]$. Therefore weak and strong nonclassical steering are easier to achieve when measuring the mode with lower purity to influence the mode with higher purity.

Finally, we should mention some similarities between our results and related works on CV quantum systems. The quantities on the left sides of (58) and (59) are well-known as the *conditional variances* appearing in the Reid EPR-criterion [66, 67], whose test is already experimentally accessible [68, 69]. This is in agreement with a result stating that quadrature measurements are the best choice for Gaussian EPR steering [70]. Expressed in these quantities, weak nonclassical steering corresponds to at least one of such variances being smaller than the vacuum value, whereas, for strong nonclassical steering, both of them have to be smaller. EPR-steerability amounts to asking that *the product* of them is smaller than the value attained by the same quantity on the vacuum [71]. More recently [72] the remote generation of Wigner negativity using bipartite Gaussian states as a starting point and one-photon subtraction instead of Gaussian measurements, has been discussed. It was found that, if the initial Gaussian state is described by a CM $\boldsymbol{\sigma}$ given in block form as in Eq. (17), then the condition for remote generation of Wigner negativity on mode $B$ by one-photon subtraction on mode $A$ is:

$$\mathrm{Tr}[\boldsymbol{\sigma}_{A|B}] \;=\; \mathrm{Tr}[\mathbf{A} - \mathbf{C}^T \mathbf{B}^{-1} \mathbf{C}] \;<\; 1 \tag{68}$$

If the initial state is in canonical form, the above inequality reduces to:

$$\left( a - \frac{c_1^2}{b} \right) + \left( a - \frac{c_2^2}{b} \right) \;<\; 1 \tag{69}$$

which is clearly implied by the strong nonclassical steering inequality from mode $B$ to mode $A$ (59), but mind the reversal of the order with respect to the generation of Wigner negativity. Therefore we can assert that two-mode Gaussian states in canonical form that are strongly nonclassically steerable are also amenable to remote generation of Wigner negativity, a stronger, non-Gaussian form of nonclassicality [73, 74] that is believed to be a necessary resource in universal quantum computation with CV systems [75, 76]. Since Ineq. (68) is not invariant under LGUTs, this conclusion cannot be directly generalized to all two-mode Gaussian states; however, in [72] the authors noted that, if one allows for passive unitary Gaussian transformations to act on mode $A$ before the one-photon subtraction, then EPR-steerability

---

[5]Note that, even in the case of a TMST state, $\mu_1 \neq \mu_A$ and $\mu_2 \neq \mu_B$ in general.

(and, a fortiori, SNS) becomes a sufficient condition for the remote generation of Wigner negativity with the most general bipartite Gaussian state as a starting point. This means that, from a resource viewpoint, *all* strongly nonclassically steerable states of two modes are suited for remote Wigner negativity generation using one-photon subtraction.

# 6   Conclusions

Upon exploring how P-nonclassicality may be generated on one mode of a TMST state by Gaussian measurements on the other mode, we have introduced, and discussed in details, the concept of nonclassical steering (NS). We have characterized all conditional states generated in this fashion by using *triangoloid plots* and we have deduced a necessary and sufficient condition for NS with TMST states, arising from the non-decreasing behaviour of the conditional nonclassicality with respect to the squeezing of the measurement. After discussing the necessity of entanglement for nonclassical steering with TMST states, and its asymmetric character, we put to use these results in the practical situation of a noisy propagation of a TWB state, for which we evaluated the maximum propagation time for NS.

We have also generalized NS to generic two-mode Gaussian states thanks to invariance under LGUTs, and two separate notions have emerged: weak and strong nonclassical steering. The first does not even imply entanglement, while the second implies EPR-steerability. We have also proved that nonclassical steering for TMST states *is* EPR steering, a conclusion that may open the way for the use of nonclassicality in QKD.

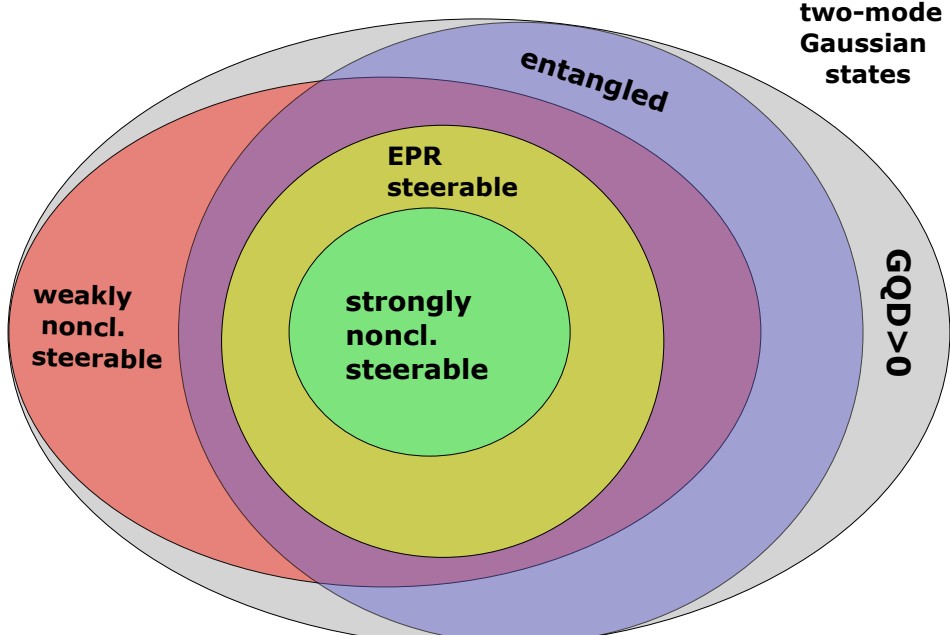

Figure 4: Quantum correlations for two-mode Gaussian states. GQD stands for Gaussian Quantum Discord. The same ordering between the modes must be adopted for all steering correlations in the diagram, while the ordering for GQD can be arbitrary.

The classification of quantum correlations for two-mode Gaussian states emerging from

our results is summarized in the diagram of Fig.4. All the quantities, with the only exception of entanglement, are asymmetric: the diagram is thus valid as long as the same ordering of the modes is chosen for weak, strong and EPR steering, while the order of Gaussian Quantum Discord (GQD) can be chosen at will. The diagram makes it clear that the sets of entangled and weakly nonclassically steerable states intersect (light-purple region and everything inside it), but neither of them is a strict subset of the other. Moreover, they are both internally tangent to the boundary of states with positive GQD, because one can find sequences of states with arbitrarily small GQD in both sets. Notice, however, that the question about a possible lower bound on GQD for states that are *both* entangled and WNS was not answered in our work and should *not* be deduced from the diagram. We also point out a possible analogy between the Gaussian steering triangoloids we have introduced and the quantum steering ellipsoids [77, 78] arising in the context of steering for a two-qubit system. Further explorations in this direction may be useful to better characterize the set of conditional states also in the CV setting.

Overall, the results of our work suggests that the hierarchy of quantum correlations for two-mode Gaussian states is more involved than previously believed and, from weakest to strongest, includes positive GQD, weak nonclassical steerability, entanglement, EPR-steerability, and strong nonclassical steerability.

# Acknowledgements

We thank C. Destri, R. Simon, A. Ferraro and M. Bondani for useful discussions. MGAP is member of INdAM-GNFM.

# A   Explicit counterexamples

## A.1   Separable states allowing weak nonclassical steering

A simple choice of parameters for the CM of a two-mode Gaussian state in canonical form, which is separable and WNS is:

$$
\begin{aligned}
a \;=\; b \;&=\; 13.9 \\
c_1 \;=\; 4.6 \,, \qquad c_2 \;&=\; -13.7
\end{aligned}
\tag{70}
$$

It is simple to check that the corresponding Gaussian state is physical ($\boldsymbol{\sigma} > 0$ and fulfilling the UR).

There are also instances of physical states in canonical form with $c_1 c_2 > 0$, a notorious sufficient condition for separability, that are nevertheless WNS. For example:

$$
\begin{aligned}
a \;=\; b \;&=\; 1.8 \\
c_1 \;=\; 0.4 \,, \qquad c_2 \;&=\; 1.6
\end{aligned}
\tag{71}
$$

Furthermore, we explicitly constructed a counterexample in terms of a Williamson's decomposition [79], [80]:

$$
\boldsymbol{\sigma}_{\text{swns}} \;=\; \boldsymbol{\Sigma}_R^{(2)} \cdot \mathbf{S}_\phi^m \cdot \left[ \boldsymbol{\Sigma}_r^{(1)} \oplus \boldsymbol{\Sigma}_r^{(1)} \right] \cdot [\boldsymbol{\sigma}_{th}(\mu_A) \oplus \boldsymbol{\sigma}_{th}(\mu_B)] \cdot \left[ \boldsymbol{\Sigma}_r^{(1)} \oplus \boldsymbol{\Sigma}_r^{(1)} \right]^T \cdot \left( \mathbf{S}_\phi^m \right)^T \cdot \left( \boldsymbol{\Sigma}_R^{(2)} \right)^T \tag{72a}
$$

$$R \;=\; \ln 2, \qquad \phi = \frac{\pi}{4}, \qquad r \;=\; \frac{1}{4}\ln\left(\frac{\mu_A + 16\mu_B}{16\mu_A + \mu_B}\right) \tag{72b}$$

where:

$$\boldsymbol{\sigma}_{th}(\mu_k) \;=\; \frac{1}{2\mu_k}\mathbb{I}_2 \tag{73a}$$

$$\mathbf{S}_\phi^m \;=\; \begin{pmatrix} \cos\phi\ \mathbb{I}_2 & \sin\phi\ \mathbb{I}_2 \\ -\sin\phi\ \mathbb{I}_2 & \cos\phi\ \mathbb{I}_2 \end{pmatrix} \tag{73b}$$

$$\boldsymbol{\Sigma}_r^{(1)} \;=\; \mathrm{diag}\left(e^{2r}, e^{-2r}\right) \tag{73c}$$

$$\boldsymbol{\Sigma}_R^{(2)} \;=\; \begin{pmatrix} \cosh R\cdot\mathbb{I}_2 & \sinh R\cdot\sigma_z \\ \sinh R\cdot\sigma_z & \cosh R\cdot\mathbb{I}_2 \end{pmatrix} \tag{73d}$$

and $\sigma_z = \mathrm{diag}(1,-1)$ is the third Pauli matrix. In physical terms, $\boldsymbol{\sigma}_{th}(\mu_k)$ is the CM of a single-mode thermal state, $\mathbf{S}_\phi^m$ performs a two-mode mixing (without cross-mixing of $x$'s and $p$'s quadratures), $\boldsymbol{\Sigma}_r^{(1)}$ implements single-mode squeezing at the level of phase-space and finally $\boldsymbol{\Sigma}_R^{(2)}$ introduces a two-mode squeezing. Notice that we choose $\phi = \frac{\pi}{4}$, so that the two-mode mixing is equivalent to the action of a balanced beam splitter, and we also took the same, real single-mode squeezing parameter $r$ for both modes. The resulting CM $\boldsymbol{\sigma}_{\mathrm{swns}}$ corresponds necessarily to a physical state, because this decomposition implies that it could be prepared with modern optical equipment, at least in principle. One can check that, for the given choice of the parameters $r, R, \phi$, the matrix $\boldsymbol{\sigma}_{\mathrm{swns}}$ is in canonical form. Moreover, for $\mu_1 = \frac{1}{32}$ and $\mu_2 = \frac{1}{4}$, it describes a Gaussian state which is both separable and weakly nonclassically steerable.

## A.2  GQD and weak nonclassical steering

Consider a sequence of Gaussian states in canonical form with the following parameters:

$$a_n \;=\; \frac{n+2}{2n+1}, \qquad b_n \;=\; n \tag{74a}$$

$$c_{1,n} \;=\; \frac{1}{\sqrt{2n}}, \qquad c_{2,n} \;=\; -\sqrt{\frac{2n}{2n+1}} \tag{74b}$$

for integers $n > 2$. By direct computation one can check that the corresponding CMs are positive and fulfilling the UR. They are also weakly nonclassically steerable, because $|c_{2,n}| > |c_{1,n}|$ and:

$$a_n - \frac{c_{2,n}^2}{b_n} \;=\; \frac{n}{2n+1} \;<\; \frac{1}{2}$$

The asymptotic values of the parameters as $n \to +\infty$ are:

$$
\begin{aligned}
a_n &\longrightarrow \frac{1}{2}^+, & b_n &\longrightarrow +\infty \\
c_{1,n} &\longrightarrow 0^+, & c_{2,n} &\longrightarrow -1^+
\end{aligned}
\tag{75}
$$

but these values have to be approached in the right way, given for example by Eq. (74), in order to respect the physical constraints for any finite $n$. As $n \to +\infty$, both the Gaussian Quantum Discords, $\mathcal{D}_{A|B}$ and $\mathcal{D}_{B|A}$, monotonically drop to zero, as we confirmed numerically. Therefore, weakly nonclassically steerable Gaussian states can have arbitrarily small GQDs *in both directions*.

### A.3    An EPR-steerable but not strongly nonclassically steerable state

We will now show that it is possible for a two-mode Gaussian state in canonical form to be EPR-steerable without being strongly nonclassically steerable. We seek parameters $a, b, c_1, c_2$ corresponding to a *physical* state, for which the greatest factor on the left-hand side of Ineq. (66) is greater than $\frac{1}{2}$ (so that the state is *not* strongly nonclassically steerable), while the other factor is small enough to ensure that the product is still smaller than $\frac{1}{4}$ (so that it *is* EPR-steerable). An instance of such a state is provided by the following choice of parameters:

$$
\begin{aligned}
a \;&=\; b \;=\; 0.9 \\
c_1 \;&=\; 0.55 \;, \qquad c_2 \;=\; -0.7
\end{aligned}
\tag{76}
$$

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
