# Peer review of "Nonclassical steering and the Gaussian steering triangoloids"

_SciPost Physics_

## Round 4 · Author Response

Reply to Referee 1

We thank the Referee for the careful reading of our manuscript and for the suggestions on how to improve the presentation of our results. Here in the following we provide a reply explaining the nature of the amendments done to the manuscript in order to comply with the points raised by Referee 1

Page 2: “causality” is a word with many potential meanings. “Signal-locality” or “no-signalling” would be unambiguous.

We agree (thanks!) with the Reviewer and have changed the sentence accordingly, also slighlty reformulating the argument for clarity.

Page 6: “vice versa” is ambiguous. “Contrapositively” or just “equivalently” would be better.

Done

Figure 4: perhaps I missed it, but I could not see where the authors proved a gap between strong nonclassical steerability and EPR-steerability. Something needs to be made clearer or commented upon. Or perhaps an additional calculation performed.

We thank the Reviewer for pointing this out. We amended the text, and added an appendix (A.3 An EPR-steerable but not strongly nonclassically steerable state) to provide an explicit example.

Reply to Referee 2

We thank the Referee for his/her careful reading of the manuscript and for his/her criticisms/remarks, which definitely helped us to correct some imprecise statements and to improve the overall presentation of our results. In the following we provide a reply to the Report of Referee 2, also detailing the nature of the changes made to the manuscript.

1) Quantum steering is by definition a non-classical feature in a larger sense than in quantum optics. Therefore, nonclassical steering sounds pleonastic to me. What the authors propose in this paper is a procedure of steering non-classicality, maybe they could find a better solution for naming this aspect of steering.

The Referee makes a good point, and we would in principle agree on a different terminology. However, since we build on the steering concept, and introduce an additional step in the hierarchy, naming our notion steering seemed a good idea. Indeed, in the last months we already presented the novel steering notion, and our results, in various forum and seminars and we found that the name is effective. It would thus be rather unfortunate to change the denomination “on the run”. For the above reasons, we would like to mantain the current title and notation. On the other hand, we have indeed employed statements involving the idea of “steering non-classicality. We also add that, strictly speaking, our notion should be referred to as P-nonclassical steering (or equivalently steering of P-nonclassicality). However, both forms seem rather redundant.

2) There is a good and organised presentation of Gaussian non-classicality defined as non-existance of the P representation as a well-behaved function. My opinion on the derivation of the "non-classicality steerability" for a TMSTS is that the authors' choice to work with purities instead of symplectic eigenvalues led to less transparent formulas, but probably this choice was related to the so-called triangoloids. However, what do Eqs. (31) tell us at first sight? To understand them, an interested reader should come back to the parameters describing the TMSTS and the measurement. Going further, the reader should discover that Eq.(34) is in fact Wiseman's steerability condition for TMSTS, Eq.(66).

Done. The sentence now reads “It is also clear, from the proof of Theorem 1 we just presented, that TMST states are EPR-steerable from one mode to the other if and only if they are nonclassically steerable in the same direction, so that the three notions of steering coincide for them and Ineq. (66) for EPR steering takes the simpler form of Ineq. (33) that we derived for nonclassical steering. ”

3) The section 5 (on the evolution of entanglement and steering of a TMSTS in local thermal reservoirs) is very interesting showing that steered non-classicality is more fragile than entanglement under thermal noise. A previous similar treatment of entanglement and discord was given in Physica Scripta 90 (2015) 074041. Eq.(56) for a squeezed vacuum state is derived and discussed there as Eq.(5.3).

We added the the paper in Physica Scripta to our list of references, and amended the text after Eq. (56) accordingly.

4) The sentence at the end of page 22 should be corrected: "there is no lower bound on two-mode squeezing that guarantees an entangled TMST state to be EPR-steerable." This lower bound exists and is written in the paper as Eq. (43) for one-way steering and can be accompanied by a similar one for the other way. As a steerability threshold, Eq.(43) was first written in Ref.[62] from an EPR-treatment.

The Referee is right if one assumes fixed values for the local purities, which is not the case in the context of the discussion after Eq. (66) (now at page 21). We however thank the Referee for pointing this out, giving us the possibility to better clarify the point. The sentence now reads ”However, the proof we provided adds quantitative aspects to these considerations: for example, it shows that, for unconstrained local purities, there is no lower bound on two-mode squeez- ing that guarantees an entangled TMST state to be EPR-steerable, as one can also see from Ineq. (41).”

---

## Round 4 · List of Changes

see authors' comments

---

## Editorial Decision

editor-in-charge_assigned